# Filtered Semi-Markov CRF

## Abstract

Semi-Markov CRF (Sarawagi and Cohen, 2005) has been proposed as an alternative to the traditional Linear Chain CRF(Lafferty et al., 2001) for text segmentation tasks such as Named Entity Recognition. In contrast to CRF, which treats text segmentation as token-level prediction, Semi-CRF considers spans as the task's basic unit, which makes it more expressive. However, Semi-CRF has two major drawbacks: (1) it has quadratic complexity over sequence length as it operates on every span of the input sequence, and (2) empirically, it performs worse than classical CRF for sequence labeling tasks such as NER. In our work, we propose Filtered Semi-Markov CRF, a Semi-CRF variant that addresses the aforementioned issues. Our model extends Semi-CRF by incorporating a filtering step for eliminating irrelevant segments, which helps reduce the complexity and dramatically reduce the search space. On a variety of NER benchmarks, we find that our approach outperforms both CRF and Semi-CRF models while being significantly faster. We will make our code available to the public.

## 1 Introduction

Sequence segmentation is the process of dividing a sequence into several distinct, non-overlapping segments to cover the entire sequence (Sarawagi and Cohen, 2005; Terzi, 2006). It has a wide range of use cases, including Named Entity Recognition (Tjong Kim Sang and De Meulder, 2003) and Chinese Word Segmentation (Li and Yuan, 1998). Sequence segmentation has traditionally been seen as a sequence labeling problem using pre-existing templates such as BIO and BILOU schemes (Ratinov and Roth, 2009). Conditional Random Field (CRF) (Lafferty et al., 2001) has been widely used in sequence labeling problems to model the dependence between adjacent token tags. Although the Linear-chain CRF has performed well in various segmentation tasks, operating at the segment level rather than the token level would be a more natural way to perform sequence segmentation. To this end, the Semi-Markov CRF (Sarawagi and Cohen, 2005) has been proposed as a variant of CRF, allowing for the incorporation of higher-level segment features, such as segment width. However, Semi-CRF, unlike CRF, is considerably slower for both learning and inference due to its quadratic complexity with respect to the sequence length. Moreover, Semi-CRF generally performs worse than CRF (sometimes the Semi-CRF performs better but the gain is only marginal) (Liang, 2005; Daumé and Marcu, 2005; Andrew, 2006). Indeed, the Semi-CRF performs joint segmentation and labeling which results in a much larger search space making learning more challenging.

To address this problem, we propose a *filtered* version of the Semi-CRF. Like Semi-CRF (Sarawagi and Cohen, 2005), our model operates on segments, but we add a filtering model to discard a large number of candidate segments. Our aim is to reduce the computational complexity by pruning the segmentation search space. During inference and after the filtering step, finding the best segmentation and labeling boils down to finding the maximum scoring path in a weighted directed acyclic graph. During training we use a similar dynamic programming algorithm allowing us to sum over all paths in the graph.

We evaluate our approach on benchmark datasets for Named Entity Recognition and find that it performs better than CRF and Semi-CRF models with noticeably faster inference. The rest of this paper is organized as follows. In the next section, we provide some background and context for understanding the foundational CRF and Semi-CRF models. Next, we present our filtered Semi-CRF model in detail, followed by the experimental setup, the results and further experimental analysis, and an overview of related works. The final section concludes this paper.

## 2 BACKGROUND

In this section, we first present the Linear-Chain CRF (Lafferty et al., 2001) and then the semi-Markov CRF (Sarawagi and Cohen, 2005), namely their structured representation and their learning and inference algorithms.

### 2.1 LINEAR CHAIN CRF

The Linear-Chain CRF (Lafferty et al., 2001) is a sequence labeling model that assigns a label to each token in the input sequence. It assumes dependencies between adjacent output labels (typically a Markov dependency of order 1). Hence, given an input sequence $\boldsymbol{x}$, a sequence of labels $\boldsymbol{y}$ of the same size $L$ is produced with $y_i \in Y$. The conditional probability of $\boldsymbol{y}$ given $\boldsymbol{x}$ is computed using the following estimator:

$$p(\boldsymbol{y}|\boldsymbol{x}) = \frac{\exp\left\{\sum_{i=1}^{L} \boldsymbol{\psi}(y_i|\boldsymbol{x}) + \sum_{i=2}^{L} \boldsymbol{T}_{y_{i-1},y_i}\right\}}{\mathcal{Z}(\boldsymbol{x})} = \frac{\exp \Psi(\boldsymbol{y}|\boldsymbol{x})}{\mathcal{Z}(\boldsymbol{x})} \tag{1}$$

where $\boldsymbol{\psi}(y_i|\boldsymbol{x}) \in \mathbb{R}$ is the score of the sequence label at position $i$ and $\boldsymbol{T} \in \mathbb{R}^{|Y| \times |Y|}$ is a learnable label transition matrix defined for each pair of labels. Furthermore, $\mathcal{Z}(\boldsymbol{x}) = \sum_{\boldsymbol{y}' \in \mathcal{Y}(\boldsymbol{x})} \exp \Psi(\boldsymbol{y}|\boldsymbol{x})$ is the partition function that serves as a normalizer of the probability distribution, where $\mathcal{Y}(\boldsymbol{x})$ is the set of all possible label sequences admissable for $\boldsymbol{x}$.

During training, the goal is to update all model parameters by minimizing the negative log probabilities of the gold labels: $-\log p(\boldsymbol{y}^*|\boldsymbol{x}) = -\Psi(\boldsymbol{y}^*|\boldsymbol{x}) + \log \mathcal{Z}(\boldsymbol{x})$. The partition function $\mathcal{Z}(\boldsymbol{x})$ is computed in polynomial time using the Forward algorithm (See Eq. 13 in Appendix A.2 for details). For inference, the goal is to produce the optimal segmentation $\boldsymbol{y}^* = \text{argmax}_{\boldsymbol{y}} \Psi(\boldsymbol{y}|\boldsymbol{x})$, which is computed using the Viterbi algorithm (Eq. 14 in Appendix A.2). The CRF has linear complexity in terms of the sequence length $L$, and quadratic complexity in terms of the number of labels $|Y|$ for both learning and inference, i.e., $O(L|Y|^2)$.

### 2.2 SEMI-MARKOV CRF

Unlike the Linear-chain CRF, the Semi-CRF (Sarawagi and Cohen, 2005) operates at the segment level to account for segment features that cannot be easily modeled using sequence labeling. The Semi-CRF produces a segmentation $\boldsymbol{y}$ (of size $M$) of input sequence $\boldsymbol{x}$ (of size $L$, with $L \geq M$). The conditional probability of the labeled segmentation $\boldsymbol{y}$ given an input $\boldsymbol{x}$ is computed as follows:

$$p(\boldsymbol{y}|\boldsymbol{x}) = \frac{\exp\left\{\sum_{k=1}^{M} \boldsymbol{\phi}(s_k|\boldsymbol{x}) + \boldsymbol{T}[l_{k-1}, l_k]\right\}}{\mathcal{Z}(\boldsymbol{x})} = \frac{\exp \Phi(\boldsymbol{y}|\boldsymbol{x})}{\mathcal{Z}(\boldsymbol{x})} \tag{2}$$

$\boldsymbol{\phi}(s_k|\boldsymbol{x}) \in \mathbb{R}$ is the score of the $k$-th segment of $\boldsymbol{y}$ and $\boldsymbol{T}[l_{k-1}, l_k]$ is the label transition score with $\boldsymbol{T}[l_0, l_1] = 0$. Furthermore, following Sarawagi and Cohen (2005), a labeled segmentation $\boldsymbol{y} = \{s_1, \ldots, s_M\} \in \mathcal{Y}(\boldsymbol{x})$ has the following properties:

- A segment $s_k = (i_k, j_k, l_k) \in \boldsymbol{y}$ consists of a start position $i_k$, an end position $j_k$, and a label $l_k \in Y$.
- The segments have positive lengths and completely cover the sequence $1 \ldots L$ *without overlapping*, i.e., $j_k$ and $i_k$ always satisfy $i_1 = 1$, $j_M = L$, $1 \leq i_k \leq j_k \leq L$, and $i_{k+1} = j_k + 1$.

For instance, for Named Entity Recognition, a segmentation of the sentence "Michael Jordan eats an apple ." would be $Y$=[(1, 2, PER), (3, 3, O), (4, 4, O), (5, 5, O), (6, 6, O)]. In (Sarawagi and Cohen, 2005), it is always assumed that non-entity segments (also O or null segments) have unit length.

The model parameters are learned to maximize the conditional probability of gold segmentation $p(\boldsymbol{y}|\boldsymbol{x})$ over the training data, similar to CRF. The partition function $\mathcal{Z}(\boldsymbol{x}) = \sum_{\boldsymbol{y}' \in \mathcal{Y}(\boldsymbol{x})} \exp \Phi(\boldsymbol{y}|\boldsymbol{x})$ can be computed in polynomial time using a modification of the Forward algorithm (Eq. 15 in Appendix A.3), and inference is done by segmental Viterbi (Eq. 16 in Appendix A.3) to produce the best segmentation $\boldsymbol{y}^* = \text{argmax}_{\boldsymbol{y}} \Phi(\boldsymbol{y}|\boldsymbol{x})$. Finally, the Semi-CRF has quadratic complexity in terms of both sequence length and the number of labels for both learning and inference, i.e., $O(L^2|Y|^2)$.

### 2.3 GRAPH-BASED FORMULATION OF SEMI-CRF

Given a sequence $\boldsymbol{x}$ of lenght $|\boldsymbol{x}| = L$, a labeled segment $s_k = (i_k, j_k, l_k)$ is defined by its start and end positions $1 \leq i_k \leq j_k \leq L$ and its label $l_k \in Y$. Let $\mathcal{G}(V, E)$ be a directed graph, whose set of nodes $V$ is made of all segments $\boldsymbol{x}$, with $|\boldsymbol{x}| = L$:

$$V = \bigcup_{i=1}^{L} \bigcup_{j=i}^{L} \bigcup_{l=1}^{|Y|} \{(i, j, l)\}, \tag{3}$$

and the directed edge $s_{k'} \to s_k \in E$ if and only if $j_{k'} + 1 = i_k$. We further define the weight of an edge $s_{k'} \to s_k$ as follows:

$$w(s_{k'} \to s_k | \boldsymbol{x}) = \phi(s_k | \boldsymbol{x}) + \boldsymbol{T}[l_{k'}, l_k] \tag{4}$$

where $\phi(s_k | \boldsymbol{x})$ is the score of the segment $s_k$ and $\boldsymbol{T}[l_{k'}, l_k]$ is the label transition score.

**Proposition 1.** *Any directed path* $\{s_1, s_2, \ldots, s_M\}$ *in the graph verifying* $i_1 = 1$ *and* $j_M = L$ *corresponds to a segmentation of* $\boldsymbol{x}$.

*Proof.* Any directed path $\{s_1, s_2, \ldots, s_M\}$ verify the properties of the segmentation described in section 2.2, *namely* $i_1 = 1$, $j_M = L$, $1 \leq i_k \leq j_k \leq L$, and $j_k + 1 = i_{k+1}$ (by **definition**). □

In addition, the score of the path $\{s_1, s_2, \ldots, s_M\}$ computed as the sum of the edge scores is equivalent to the Semi-CRF score (2.2) of the segmentation $\boldsymbol{y} = \{s_1, s_2, \ldots, s_M\}$:

$$\begin{aligned} \texttt{score}(s_1, s_2, \ldots, s_M) = \sum_{k=1}^{M} w(s_{k-1} \to s_k | \boldsymbol{x}) = \sum_{k=1}^{M} \phi(s_k | \boldsymbol{x}) + \boldsymbol{T}[l_{k-1}, l_k] \\ = \Phi(\boldsymbol{y} = \{s_1, \ldots, s_M\} | \boldsymbol{x}) \end{aligned} \tag{5}$$

The search for the best segmentation consists in finding the maximal weighted path of the graph that begins at $i_1 = 1$ and end at $j_M = L$. Finding the best path in this graph has a complexity of $L^3$ using a generic search algorithm such as Bellman-Ford (see section 3.3 for details). Nevertheless, taking into account the lattice structure of the problem allows reducing the complexity to $L^2$, as is done in the Viterbi algorithm (Viterbi, 1967).

## 3 FILTERED SEMI-MARKOV CRF

We describe in this section our proposed alternative to Semi-CRF, which we term Filtered Semi-CRF. The motivations for this new model is to address two weaknesses of the Semi-CRF. First, the Semi-CRF is not well-suited for long texts due to its quadratic complexity and the search space is prohibitively large. Second, in tasks such as NER where some segments should be labeled `null`, multiple paths in the Semi-CRF graph can produce the same set of entities. This is because long `null` segments can be broken into smaller contiguous `null` segments without modifying the result. In fact, Sarawagi and Cohen (2005) constrains `null` segments to have a unit length and assigns them a score. The crux of our approach is to use an independent model to filter the Semi-CRF graph described in § 2.3 prior to further computations. The resulting graph is order of magnitude smaller than the original one and does not contain `null` segments thus addressing both issues.

### 3.1 FILTERING

In our model, filtering is applied to the full set of segments (we denote as $V_{full}$). The filtering eliminates the segments that are predicted to be `null` segments by means of a local classifier $\phi_{local}$ :

$$V = \left\{ s_k \in V_{full} \mid \arg\max_{l_k} \phi_{local}(s_k = (i_k, j_k, l_k) | \boldsymbol{x}) \neq \texttt{null} \right\} \tag{6}$$

Since the filtered nodes $V$ may not contain all segments, defining the edges $E$ as we did in 2.3 would not be applicable here. Thus, we propose to define the edges using the method of Liang et al.

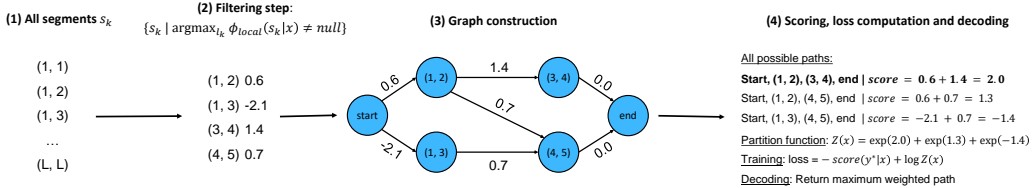

Figure 1: **Filtered Semi-Markov CRF**. 1) Enumerate all the segments of the input sequence. 2) `null` segments are dropped (Eq. 6) using a local segment classifier $\phi_{local}$. 3) Construct the path graph from the filtered segments; we omit the transition scores for better readability. 4) During training, we compute the loss function (Eq. 9 and 10) by constraining the gold path $y^*$ to be a path of the graph, and during inference, return the maximum weighted path (Alg. 2). Please note that the size of the graph can vary a lot depending on the input sequence and training stage (Fig. 2 and 3).

(1991): $\forall (s_{k'}, s_k) \in V^2$, $s_{k'} \to s_k \in E$ if $j_{k'} < i_k$ and there is no $s_{k^*} \in V$ such that $j_{k'} < i_{k^*}$ and $i_{k^*} < j_k$. This formulation means that $s_{k'} \to s_k$ is an edge if $s_k$ begins after $s_{k'}$, and that no other segment lies completely inside $(j_{k'}, i_k)$. This formulation generalizes the Semi-CRF to graphs with missing segments. However, when segments are missing, the starting and ending of segmentation are not necessarily $i_1 = 1$ and $j_M = L$. To fix this problem, we simply add two terminal nodes `start` and `end`:

- `start` $\to s_k \in E$ if $s_{k'} \to s_k \notin E$ for all $k' \neq$ `start`
- $s_k \to$ `end` $\in E$ if $s_k \to s_{k'} \notin E$ for all $k' \neq$ `end`

A segmentation in the graph is a path from `start` to `end`, i.e., $\{s_0, s_1, \ldots, s_M, s_{M+1}\}$ with $s_0 =$ `start` and $s_{M+1} =$ `end`.[1] An illustration of the graph construction is shown in the figure 1.

For named entity recognition, if we take again the example of Section 2.2, the correct segmentation of "Michael Jordan eats an apple." using the Filtered Semi-CRF would be $\boldsymbol{y}$=[`start`, $(1, 2, \text{PER})$, `end`], the remaining segments being considered as `null` label: in fact, the Filtered Semi-CRF only accounts for entity segments and assumes that the remaining parts of the sequence have the `null` label.

### 3.2 Scoring, Learning and Inference

**Segmentation probability** To compute a segmentation score in the filtered graph, we sum the weights of the path edges representing the segmentation as for the Semi-CRF described in Section 2.3:

$$\text{score}(\boldsymbol{y} = \{s_0, \ldots, s_{M+1}\}|\boldsymbol{x}) = \sum_{s_k \in \boldsymbol{y}} w(s_{k-1} \to s_k | \boldsymbol{x}) \tag{7}$$

where $w(s_{k-1} \to s_k | \boldsymbol{x}) = \phi_{global}(s_k | \boldsymbol{x}) + \boldsymbol{T}[l_{k-1}, l_k]$ if $k \notin \{1, M+1\}$ and $w(s_0 \to s_1) = \phi_{global}(s_1 | \boldsymbol{x})$ and $w(s_M \to s_{M+1}) = 0$, where $s_0 =$ `start` and $s_{M+1} =$ `end`. Note that the `start` and `end` nodes are added only to make the problem a single-source, single-destination shortest path problem. Moreover, $\phi_{global}$ is a neural network, similarly to $\phi_{local}$, it takes as input the labeled segments $s_k = (i_k, j_k, l_k) \in V$ and returns their scores. Finally, the segmentation probability of the Filtered Semi-CRF is:

$$p(\boldsymbol{y} = \{s_0, \ldots, s_{M+1}\}|\boldsymbol{x}) = \frac{\exp \text{score}(\boldsymbol{y}|\boldsymbol{x})}{\mathcal{Z}(\boldsymbol{x})} \tag{8}$$

$\mathcal{Z}(\boldsymbol{x}) = \sum_{\boldsymbol{y}' \in \mathcal{Y}(x)} \exp \text{score}(\boldsymbol{y}'|\boldsymbol{x})$ is the partition function, which makes the probabilities of all segmentation sum to one.

---

[1]It is worth noting that the segmentation problem can be formulated as finding a the highest scoring Maximal Independent Set (*MIS*) in a *interval graph* (Gupta et al., 1982).

The set $\mathcal{Y}(x)$ contains all paths in the graph from start to end. For a reasonably small graphs, $\mathcal{Y}(x)$ can be enumerated, but this is intractable for larger graphs. The partition function can be efficiently computed without enumeration; with dynamic programming using a variant of the Bellman-Ford algorithm, which can be seen as a message-passing algorithm (Wainwright and Jordan, 2008):

---

**Algorithm 1** Computing $\mathcal{Z}(\boldsymbol{x})$

---

1: Topologically sort the nodes of $V$
2: $\alpha[\text{start}] = 1$ and $\alpha[k] = 0$ otherwise **for** $k \in V$
3: **for all** $k \neq \text{start}$ in $V$ **do**
4:      **for all** $k'$ such that $k' \to k \in E$ **do**
5:         $\alpha[k] \leftarrow \alpha[k] + \alpha[k'] \exp\{w(s_{k'} \to s_k)|\boldsymbol{x}\}$
6:      **end for**
7: **end for**
8: $\mathcal{Z}(\boldsymbol{x}) = \alpha[\text{end}]$

---

In practice, this implementation of $\mathcal{Z}(\boldsymbol{x})$ is unstable, so we did all the computations in the log space to prevent overflow/underflow. The complexity of the algorithm is $O(|V| + |E|)$. We provide more details about the size of $V$ and $E$ as a function of $L$ in Section 3.3.

**Learning** During training, we jointly minimize the filtering loss and the segmentation loss. The filtering loss $\mathcal{L}_{local}$ of the local classifier $\phi_{local}$ is the sum of the negative log-probability of all gold labeled segments of the training set $\mathcal{T}$. Since the filtering task is highly imbalanced, we down-weight the loss for the label $l = \text{null}$ as a mean of regularization. The weighting ratio $\beta \in [0, 1]$ is tuned on the development set:

$$\mathcal{L}_{local} = - \sum_{\substack{(i,j,l)\in\mathcal{T} \\ l\neq\text{null}}} \log p(i,j,l|\boldsymbol{x}) - \beta \times \sum_{\substack{(i,j,l)\in\mathcal{T} \\ l=\text{null}}} \log p(i,j,l|\boldsymbol{x}) \tag{9}$$

where $p(i, j, l|\boldsymbol{x})$ is the probability that the segment $(i, j)$ has the label $l$ using the local classifier $\phi_{local}$. The loss of the segmentation model $\phi_{global}$ is computed as:

$$\mathcal{L}_{global} = -\text{score}(\boldsymbol{y}|\boldsymbol{x}) + \log \mathcal{Z}(\boldsymbol{x}) \tag{10}$$

Furthermore, during the training, we constrain the candidate segments $V$ (Eq. 6) to contain the gold entity segments $\boldsymbol{y}$, and we also ensure that the gold segmentation is a path of the filtered graph, i.e., all other candidate spans should be overlapping at least with one segment of the gold. This choice may be sub-optimal since it can cause exposure bias, i.e., a training-inference discrepancy. However, we found that it works well in practice, and suppressing it leads to unstable learning and a negative value of the global loss since $\text{score}(\boldsymbol{y}|\boldsymbol{x})$ can be larger than $\log \mathcal{Z}(\boldsymbol{x})$. The total loss of the model is the sum of the local and global losses, $\mathcal{L}_{total} = \mathcal{L}_{global} + \mathcal{L}_{local}$.

**Inference** During inference, the objective is to return the path (from start to end) of the graph that has the best score. We solve this problem using a max-sum dynamic programming algorithm that has the same structure as Algorithm 1:

---

**Algorithm 2** Decoding

---

1: Topologically sort the nodes of $V$
2: $\delta[\text{start}] = 0$
3: **for all** $k \neq \text{start}$ in $V$ **do**
4:
$$\delta[k] = \max_{\substack{k' \\ (k' \to k)\in E}} \delta[k'] + w(s_{k'} \to s_k|\boldsymbol{x})$$

5: **end for**

---

The highest scoring path, i.e $\text{argmax}_{\boldsymbol{y}}\text{score}(\boldsymbol{y}|\boldsymbol{x})$, is the path traced by $\delta[\text{end}]$ which can be obtained by backtracking. This algorithm has a complexity of $O(|V| + |E|)$, the same as computing the partition function $\mathcal{Z}(\boldsymbol{x})$.

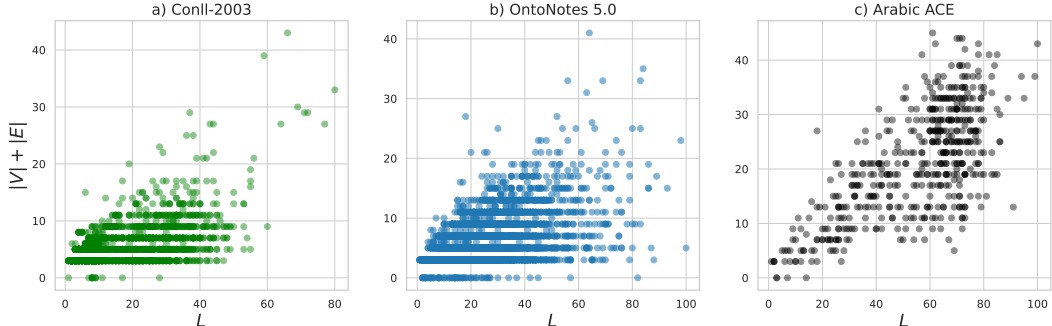

Figure 2: **Empirical complexity analysis**. This plot illustrates the relationship between the size of the filtered graph ($|V| + |E|$) and the input sequence length $L$, on three NER datasets. This experment is done with trained models.

### 3.3 COMPLEXITY ANALYSIS

In this section, we analyze the complexity of the algorithms (1 and 2) $O(|V| + |E|)$ as a function of the input sequence length $L$. Note that the size of $V$ does not depend on the number of labels $|Y|$ since there is at most one label per segment due to the filtering step in equation 6.

**Proposition 2.** *There are $\frac{L(L+1)}{2}$ nodes in a complete segment path graph constructed using a sequence of length L.*

**Proposition 3.** *There are $\frac{L(L-1)(L+1)}{6}$ edges in a complete segment path graph constructed from a sequence of length L.*

We use propositions 2 and 3 to derive the complexity of the Filtered Semi-CRF model, developed below. Their proofs can be found in Appendix A.1.

**Worst case complexity**    In the worst case, the filtering model $\phi_{local}$ does not filter any segments, i.e., all segments are kept. From propositions 2 and 3, we can deduce that in the worst case, $O(|V|) = O(L^2)$ and $O(|E|) = O(L^3)$ which means that the complexity of our worst case algorithm is cubic as a function of the sequence length $L$ since $O(|V| + |E|) = O(L^3)$. However, note that in the worst case, the resulting graph is the Semi-CRF and the complexity can be reduced to $L^2$ using the algorithms Forward (during training) and Viterbi (during inference).

**Best case complexity**    The best case scenario means that the filtering is perfect, so the number of nodes in the graph $|V|$ is equal to the true number of non-`null` segments in the input sequence, which we denote by $\mathcal{J}$. Moreover, since $\mathcal{J}$ does not contain overlapping segments, $|\mathcal{J}| \leq L$ with $|\mathcal{J}| = L$ if all segments in $\mathcal{J}$ have unit length and cover the entire sequence i.e $\mathcal{J} = \{(i, i, l_i)|i = 1 \ldots L, l_i \neq \texttt{null}\}$. Furthermore, $|E| = |\mathcal{J}| - 1 \leq L - 1$ because perfect filtering implies that the path number is unique. Finally, we can conclude that the complexity is linear i.e, $O(|V| + |E|) = O(L)$.

**Empirical analysis**    We further investigate the empirical complexity of our approach by looking for a relationship between $|V| + |E|$ and the sequence length $L$ in practice. We performed the experiments on three text segmentation datasets, *Conll-2003*, *OntoNotes 5.0* and *Arabic ACE* dedicated to the task of Named Entity Recognition. The results are shown in Figure 2. The plots show that the graph size $|V| + |E|$ is generally smaller than the sequence length $L$ for a trained model, meaning that empirically, the complexity is close to the best case complexity which is $O(L)$. However, during training, especially in the first stage, the size of the graph can be large because the filtering model may be poor, as illustrated in the figure 3. Empirically, the early steps of the training can be time consuming due to larger graph size. However, after a few gradient steps, the size of the graph decreases significantly since most of the segments of an input sequence are labeled as `null`.

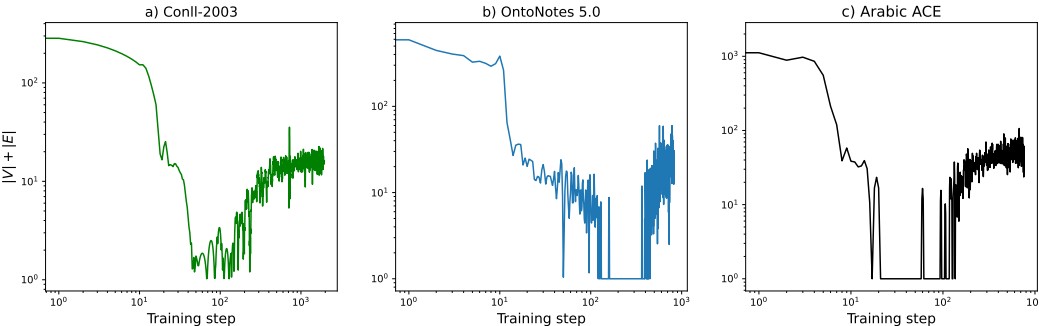

Figure 3: **Evolution of the graph size during training**. The two axis are in log-scale and the data are smoothed using Savitzky–Golay filter (Savitzky and Golay, 1964). There are three main stages. At the beginning of the training, the size of the graph is large because the filtering model is not trained. At the second stage, the size of the graph is small because the filtering model is confident about `null` segments (most segments are null). At the last stage, the size of the graph is stabilizing.

## 4 EXPERIMENTAL SETUPS

### 4.1 REPRESENTATION AND SCORES

For all our models, we used pre-trained transformer models (Devlin et al., 2019) to compute word representations. Specifically, the input sequence $\{x\}_{i=1}^n$ is fed into a pre-trained transformer producing a set of contextualized embeddings $\{h\}_{i=1}^n \in \mathbb{R}^D$, with $D$ the embedding size of the model. In addition, since pre-trained transformers typically separate words into sub-tokens, we use the first sub-token embedding as the representation of the whole word, which is a common practice for token-level prediction tasks.

**Token scores** In our sequence labeling baseline, we compute the label score at position $i$ as a linear projection of the token representation at the same position: $\boldsymbol{\psi}(y_i|\boldsymbol{x}) = \boldsymbol{w}_y^T \boldsymbol{h}_i \in \mathbb{R}$, where $\boldsymbol{w}_y \in \mathbb{R}^{D \times 1}$ is a label-specific learnable weight vector.

**Segment scores** For our segment-level models (Semi-CRF and FSemiCRF), we compute the representation $\boldsymbol{s}_{i:j}$ of the segment $(i, j)$ using a sum pooling of the representations of the tokens comprising the segment, $\boldsymbol{s}_{i:j} = \text{SUM}([\boldsymbol{h}_i, \boldsymbol{h}_{i+1}, \ldots, \boldsymbol{h}_j])$.

Indeed, according to Adi et al. (2017), sum pooling can effectively model the length of the sequence. Moreover, for the segment-based models (i.e, Semi-CRF and Filtered Semi-CRF), we restrict the segment to a maximum width to reduce complexity without harming the recall score on the training set (however some segments may be missed for the test set). By bounding the maximum width of the segments, we reduce the number of segments from $L^2$ to $LK$, where $K$ is the maximum width. Thus, under this setup, the the complexity of the Semi-Markov CRF become $O(LK|Y|^2)$. Finally, the segment scores (i.e all $\phi_.(s_k)$) are computed using a linear projection of the segment representations, analogous to token scores.

### 4.2 SETUP

**Datasets and evaluation** We evaluate our models on three diverse datasets of Named Entity Recognition. *conll-2003* (Tjong Kim Sang and De Meulder, 2003) is a dataset from the news domain designed for extracting entities such as Person, Location, and Organisation. *OntoNotes 5.0* (Weischedel et al., 2013) is a large corpus comprising various kinds of text, including newswire, broadcast news, and telephone conversation, with a total of 18 different entity types, such as Person, Organization, Location, Product, or Date. *Arabic ACE* is the Arabic portion of the multilingual information extraction corpus, ACE 2005 (Walker et al., 2006). It includes texts from a wide range of genres, such as newswire, broadcast news, and weblogs, with a total of 7 entity types. We follow the standard common approach for evaluating NER models, based on exact matching between predicted and gold entities, discarding non-entity segments. We report the micro-averaged precision (P), recall (R), and the F1-score (F) on the test set for models selected on the dev set.

| Models | Conll-2003 | | | OntoNotes 5.0 | | | Arabic ACE | | |
|---|---|---|---|---|---|---|---|---|---|
| | P | R | F | P | R | F | P | R | F |
| (Yu et al., 2020) | 93.7 | 93.3 | 93.5 | 91.1 | 91.5 | 91.3 | | | |
| (Yan et al., 2021) | 92.61 | **93.87** | 93.24 | 89.99 | 90.77 | 90.38 | | | |
| (Zhu and Li, 2022) | 93.61 | 93.68 | 93.65 | **91.75** | **91.74** | **91.74** | | | |
| (Shen et al., 2022) | 93.29 | 92.46 | 92.87 | 91.43 | 90.73 | 90.96 | | | |
| Our experiments | | | | | | | | | |
| CRF | 93.29 | 92.21 | 92.75 | 89.00 | 90.16 | 89.57 | 82.79 | 84.44 | 83.61 |
| Semi-CRF | 92.37 | 90.49 | 91.42 | 88.91 | 89.78 | 89.34 | 82.97 | 84.24 | 83.60 |
| + Unit size `null`[†] | 92.08 | 91.41 | 91.74 | 89.17 | 89.76 | 89.47 | 83.35 | 83.62 | 83.48 |
| FSemiCRF | **94.72** | 93.09 | **93.89** | 90.69 | 91.31 | 91.00 | 83.43 | **85.51** | **84.46** |
| – w/o $\mathcal{L}_{global}$ (10)[†] | 94.24 | 92.70 | 93.46 | 90.85 | 89.57 | 90.21 | **83.73** | 83.56 | 83.64 |

Table 1: **Main results**. All English models employ `bert-large-cased` for representing the tokens on English datasets, except (Yan et al., 2021) that uses bart-large. [†] See ablation study (sec. 5.2) for details about these models.

**Hyperparameters**   To produce contextual token representations, we used `bert-large-cased` (Devlin et al., 2019) for both *conll-2003* and *OntoNotes 5.0* datasets, and `bert-base-arabertv2` (Antoun et al., 2020) for *Arabic ACE*. For simplicity, we do not use auxiliary embeddings (eg. *character embeddings*). All models are trained with *Adam* optimizer (Kingma and Ba, 2017). We employed a learning rate of `2e-5` for the pre-trained parameters and a learning rate of `5e-4` for the other parameters. We used a batch size of 8 and trained for a maximal epoch of 15. We keep the best model on the validation set for testing. We trained all the models on a server equipped with V100 GPUs. We implemented our model with PyTorch (Paszke et al., 2019). The pre-trained transformer models were loaded from HuggingFace's Transformers library (Wolf et al., 2019). We used AllenNLP (Gardner et al., 2018) for data preprocessing and the seqeval library (Nakayama, 2018) for evaluating the sequence labeling models. Our Semi-CRF implementation is based on pytorch-struct (Rush, 2020).

**Baselines**   We compare our Filtered Semi-CRF against the CRF (Lafferty et al., 2001) and the Semi-CRF (Sarawagi and Cohen, 2005). We also report some results from the literature: Bi-affineNER (Yu et al., 2020), Bart-NER (Yan et al., 2021), Boundary Smoothing (Zhu and Li, 2022) and PIQN (Shen et al., 2022). For English datasets, all the models are using `bert-large-case` for token representation except BartNER which used `bart-large` (Lewis et al., 2020). Moreover, for a fair comparison, we only report results for models using sentence-level context (in contrast to paragraph-level context).

## 5   RESULTS

### 5.1   MAIN RESULTS

**CRF v.s. Semi-CRF v.s. FSemiCRF**   We here compare our proposed model to the CRF and Semi-CRF baseline models reported in Table 1. Semi-CRF is the worst-performing model, with the lowest scores on conll-2003 and *OntoNotes 5.0* datasets and the same performance as CRF on the *Arabic ACE* dataset. Moreover, on all datasets, our proposed FSemiCRF outperforms CRF and Semi-CRF in terms of precision and recall, demonstrating its utility in a variety of scenarios. Furthermore, we find that there is no significant difference between FSemiCRF with and without transition scores (in fact, most of the time the result is the same), which can be explained by the fact that adjacent segments in the filtered graph may be far from each other.

**Comparison to SOTA**   Compared to the state-of-the-art models, our FSemiCRF has the highest score on the *Conll-2003* dataset, outperforming the second-highest score by 0.24 in terms of F1-score. On OntoNotes, while not the best, our model is still competitive.

| Datasets | $|Y|$ | Training | | | Inference | | |
|---|---|---|---|---|---|---|---|
| | | CRF | SemiCRF | FSemiCRF | CRF | SemiCRF | FSemiCRF |
| *conll-2003* | 4 | **9.51** | 8.73 | *7.50* | *20.41* | 17.35 | **31.54** |
| *OntoNotes 5.0* | 18 | **7.68** | 3.38 | *5.49* | *13.63* | 4.26 | **23.22** |
| *Arabic ACE* | 7 | *4.60* | 4.14 | **6.12** | *7.87* | 6.30 | **12.60** |

Table 2: **Model Throughput** (higher is better). We measure the throughput of the model in *batch per second*, using a batch size of 8 on a V100 GPU. All models use the same vector dimensions for token representation for fair comparison.

## 5.2 ABLATION STUDY

**Semi-CRF + Unit size `null`** We study an alternative variant of the Semi-CRF that allows `null` labels only for segments of unit length. To do this, we simply modify the original Semi-CRF by eliminating/masking segmentation paths that contain null segments whose size is greater than one. The motivation for this study is to reduce the search space and force out segmentation ambiguity. We can see that it improves the results on *conll-2003* and *OntoNotes 5.0*. However, the results are still poor compared to the other approaches.

**FSemiCRF w/o global loss** As shown in Table 1, we investigate the influence of global loss on FSemiCRF by removing it, resulting in a local span-based NER model. Its decoding is performed using a greedy algorithm where the highest-scoring entity is iteratively added to the result as long as it does not overlap with the previously selected entities. As shown in the Table 1, even without the global loss, the model is competitive, but the global model consistently improves the scores.

## 5.3 EFFICIENCY ANALYSIS

In this section, we analyse the computational efficiency of the models both for training and inference. We performed two experiments: 1) the training and inference throughput in Table 2, measured in *batch per second*; 2) the inference wall clock time for comparing the Semi-CRF and FSemiCRF to show the time needed for computing the span scores and the decoding, in *millisecond per sample*. For both experiments, we use a batch size of 8 and an Nvidia V100 GPU with 16 GB of memory. For a fair comparison, for all the datasets and models, we employed a similar model size for the token representation, namely `bert-base-cased` for *Conll-2003* and *OntoNotes 5.0* and `bert-base-arabertv2` for *Arabic ACE*.

**Throughput** For training, the results show that the CRF model is the fastest for most of the datasets. Then, the FSemiCRF is the second fastest; it has a better training throughput than the Semi-CRF on all datasets except for *Conll-2003*. We empirically found that the speed of the Semi-CRF depends strongly on the number of labels; therefore, it is fast on *Conll-2003* since this dataset has only a few label types. During inference, our FSemiCRF is significantly faster than other methods: it is 5 times faster than Semi-CRF on *OntoNotes 5.0* and 2 times faster on *Arabic ACE*. We explain this behavior by two main points: 1) During inference, segment filtering is highly parallelizable, while during training it is not. 2) The complexity of FSemiCRF strongly depends on the performance of the filtering model; at the early stage of training, the filtering model may be poor, which leads to a larger graph (as shown in the figure 3) while the size of the graph is generally small during inference. See section 3.3 for more detail.

**Wall clock time** We performed a wall clock time analysis of the Semi-CRF and Filtered Semi-CRF on the table 3. As shown in the table, computing the segment scores (using bert-based models) is the same for both approaches. However, for the decoding, Semi-CRF applies the segmental Viterbi algorithm to the segments, while FSemiCRF only uses the filtered segments. This study shows that the decoding time of the FSemiCRF is almost negligible compared to computing the segment scores. In contrast, the decoding for Semi-CRF is significantly slower. Noticeably, the decoding is sometimes slower than computing the segment score for the Semi-CRF, which is the case on *OntoNotes 5.0* and *Arabic ACE* datasets.

| | Conll-2003 | | OntoNotes 5.0 | | Arabic ACE | |
|---|---|---|---|---|---|---|
| | Semi-CRF | FSemiCRF | Semi-CRF | FSemiCRF | Semi-CRF | FSemiCRF |
| Scoring | 3.85 ms | | 4.88 ms | | 8.33 ms | |
| Decoding | 3.71 ms | 0.21 ms | 27.5 ms | 0.22 ms | 10.13 ms | 0.33 ms |
| Total | 7.56 ms | 4.06 ms | 32.38 ms | 5.10 ms | 18.47 ms | 8.66 ms |

Table 3: **Wall clock time** (lower is better). This table reports the average wall-clock time comparison of Semi-CRF and Filtered Semi-CRF in milliseconds (per sample). We separate the time needed for computing the segment representations (with BERT) and the decoding algorithm. Please note that the scoring time is the same for Semi-CRF and FSemiCRF. We use the same setup as in table 2.

## 6 RELATED WORK

Many frameworks have been proposed for text segmentation. The most popular is the Linear-Chain CRF (Lafferty et al., 2001), which treats text segmentation tasks as token-level prediction. It is trained by maximizing the sequence-level objective of the gold standard labeling and using the Viterbi algorithm (Viterbi, 1967; Forney, 2010) for decoding, adding some constraints to the transition matrix to enforce the well-formedness of the output. First variants employed handcrafted features (Lafferty et al., 2001; Gross et al., 2006; Roth and tau Yih, 2005) and it has been further extended to automatic feature learning using neural networks (Do and Artières, 2010; van der Maaten et al., 2011; Kim et al., 2015; Huang et al., 2015; Lample et al., 2016). Usually, CRF is used with a 1st order Markov transition on the labels, but other methods such as Ye et al. (2009) and Cuong et al. (2014) have proposed to employ higher order dependency to further enhance the performance, however due to the high complexity and the marginal gains, it has not gained in popularity. Semi-CRF (Sarawagi and Cohen, 2005) has been proposed as an alternative to CRF for sequence segmentation tasks. Instead of operating on the token level, the Semi-CRF considers segments as the basic unit for the prediction. It has been applied to several sequence segmentation tasks, such as Chinese word segmentation (Kong et al., 2016) and Named Entity Recognition (Sarawagi and Cohen, 2005; Andrew, 2006; Zhuo et al., 2016; Liu et al., 2016; Ye and Ling, 2018). Its main advantage over traditional CRFs is that it can incorporate segment-level features such as segment length, which can help obtain a model with higher predictive ability. However, it presents two major shortcomings: it has quadratic complexity as a function of the sequence length, which makes it difficult to apply for long sequences, and it generally obtains inferior or marginal gains over the CRFs (Liang, 2005; Daumé and Marcu, 2005; Andrew, 2006). In this work, we proposed a more efficient alternative by adding a filtering step that drops `null` segments. Our approach provides significantly better performance and more efficient inference than both CRF and Semi-CRF.

## 7 CONCLUSION

In this paper, we proposed Filtered Semi-CRF, a novel technique for text segmentation tasks. We applied our method to the Named Entity Recognition (NER) task and obtained significant gain over traditional CRF and Semi-CRF models on various benchmark datasets. In addition to being more efficient, our algorithm is faster and more scalable than the baseline models. In future work, we plan to extend our algorithm to nested segment structures.

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

## A  APPENDIX

### A.1  PROOFS

**Proposition 2.** *There are $\frac{L(L+1)}{2}$ nodes in a complete segment path graph constructed using a sequence of length L.*

*Proof.* Nodes are the enumeration of all segments (regardless of labels). Thus,

$$V = \bigcup_{i=1}^{L}\bigcup_{j=i}^{L}(i,j) \implies |V| = \sum_{i=1}^{L}\sum_{j=i}^{L}1 = \sum_{i=1}^{L}(L+1-i)$$

$$= \sum_{i=1}^{L}(L+1) - \sum_{i=1}^{L}i = L(L+1) - \frac{L(L+1)}{2} \tag{11}$$

$$|V| = \frac{L(L+1)}{2}$$

$\square$

**Proposition 3.** *There are $\frac{L(L-1)(L+1)}{6}$ edges in a complete segment path graph constructed from a sequence of length L.*

*Proof.* We know that in the complete segment graph

1. By definition, $(i_k, j_k) \to (i_{k'}, j_{k'}) \in E$ iff $j_k + 1 = i_{k'}$

2. There are $j_k$ segments ending at $j_k$ i.e $|\bigcup_{i=1}^{j_k}(i, j_k)| = j_k$

3. There are $L - j_k$ segments starting at $i_{k'}$ i.e $|\bigcup_{i=i_{k'}}^{L}(i_{k'}, i)| = L - i_{k'} + 1 = L - j_k$

From 1, 2 and 3, we can deduce that there is $j_k(L - j_k)$ segments starting at $i_{k'}$ and ending at $j_k$. Finally, the total number of edges of the graph is the sum over all $j_k$ from 0 to $L$:

$$|E| = \sum_{j_k=1}^{L} j_k(L - j_k) = L\sum_{j_k=1}^{L} j_k - \sum_{j_k=1}^{L} j_k^2$$

$$= L\frac{L(L+1)}{2} - \frac{L(L+1)(2L+1)}{6} = L(L+1)(\frac{L}{2} - \frac{2L+1}{6}) \tag{12}$$

$$|E| = \frac{L(L+1)(L-1)}{6}$$

$\square$

### A.2  CRF

**Partition function**  The partition function $\mathcal{Z}(\boldsymbol{x})$ of the CRF (Lafferty et al., 2001) is computed using the forward algorithm, with $\alpha(1, y) = \boldsymbol{\psi}(y|\boldsymbol{x})$ and for $i = 2 \dots L$:

$$\alpha(i, y) = \sum_{y' \in Y} \alpha(i-1, y')\exp\{\boldsymbol{\psi}(y|\boldsymbol{x}) + \boldsymbol{T}_{y',y}\}$$

$$\mathcal{Z}(\boldsymbol{x}) = \sum_{y \in Y} \alpha(L, y) \tag{13}$$

**Decoding**  The decoding of CRF is done with the Viterbi algorithm, with $\delta(1, y) = \boldsymbol{\psi}(y|\boldsymbol{x})$

$$\delta(i, y) = \max_{y' \in Y} \delta(i-1, y') + \boldsymbol{\psi}(y|\boldsymbol{x}) + \boldsymbol{T}_{y',y} \tag{14}$$

The best labeling is given by the path traced by $\max_{y \in Y} \delta(L, y)$. Both the computation of the partition function and the decoding of the CRF have a complexity of $O(L|Y|^2)$.

## A.3 SEMI-CRF

**Partition function**   The partition function of the Semi-CRF (Sarawagi and Cohen, 2005) $\mathcal{Z}(\boldsymbol{x})$ is computed using the following dynamic program (a modification of the forward algorithm) with $\alpha(0,:) = 1$ and $\alpha(m,:) = 0$ if $m < 0$ and otherwise:

$$\alpha(m,y) = \sum_{d=1}^{L} \sum_{y' \in Y} \alpha(m-d,y') \exp\left\{\boldsymbol{\phi}((i=m-d+1, j=m, l=y)|\boldsymbol{x}) + \boldsymbol{T}[y',y]\right\} \tag{15}$$

$$\mathcal{Z}(\boldsymbol{x}) = \sum_{y \in Y} \alpha(L,y)$$

**Decoding**   The decoding of the Semi-CRF is done with the segmental/Semi-Markov Viterbi algorithm with $\delta(0,:) = 0$ and $\delta(m,:) = -\infty$ if $m < 0$ and otherwise:

$$\delta(m,y) = \max_{\substack{y' \in Y \\ d=1\ldots L}} \delta(i-d,y') + \boldsymbol{\phi}((i=m-d+1, j=m, l=y)|\boldsymbol{x}) + \boldsymbol{T}[y',y] \tag{16}$$

The highest scoring segmentation is the path traced by $\max_{y \in Y} \delta(L,y)$. Both the computation of the partition function and the decoding of the Semi-CRF have a complexity of $O(L^2|Y|^2)$.

