# OpenReview forum: "Filtered Semi-Markov CRF"
_ICLR.cc/2023/Conference — Submitted to ICLR 2023_

### Official Review · Reviewer_YFd7 · 2022-10-25

**Confidence:** 5
**Correctness:** 4
**Technical Novelty And Significance:** 2
**Empirical Novelty And Significance:** 2
**Recommendation:** 3

**Clarity, Quality, Novelty And Reproducibility:**

Clarity: mostly high
Quality: a bit weak
Novelty: probably new
Reproducibility: good

**Strength And Weaknesses:**

The paper is clearly written. The method is, to my knowledge, original.

It does not compare to state-of-the-art methods, which generally use something like transformer decoders with greedy decoding. (https://paperswithcode.com/sota/named-entity-recognition-ner-on-conll-2003, https://paperswithcode.com/sota/named-entity-recognition-ner-on-ontonotes-v5, https://paperswithcode.com/sota/named-entity-recognition-on-ace-2005) The results are a bit worse than SOTA.

The motivation is to address the shortcoming of semiCRF that it considers all spans. It turns out that the proposed method is not only faster but gets better evaluation scores than pure semi-CRF. (The authors might want to speculate on why that is-- it isn't clear to me a priori that the proposed method should be more accurate.) However the proposed method has some shortcomings: it introduces a hyperparameter beta, it has the possibility of discarding good spans in the filtering phase, and the worst-case complexity is higher. For many tasks it is possible to give a reasonable upper bound on the span length that needs to be considered, so complexity of semiCRF training/inference is reduced to linear from quadratic in sentence length. What about a baseline that uses a maximum segment length and allows null labels only for spans of length 1? I think that should address the weaknesses of semiCRF in a cleaner way and I would not be surprised to see it working quite well in terms of evaluation.

**Summary Of The Paper:**

A method is proposed for filtering spans to be labeled in a semi-Markov CRF, which empirically reduces training/inference complexity and increases evaluation metrics relative to basic CRF or basic semi-CRF.

**Summary Of The Review:**

The proposed method does well in ablation experiments (reducing to CRF, or expanding to full semi-CRF) but some obvious baselines were neglected (greedy decoding over Transformer, bounding maximum span of semiCRF).

---

> ### Author Response · Authors · 2022-11-18
> **Response to Reviewer YFd7**
>
> Thank you for your comment. We address your concerns below.
>
> > It does not compare to state-of-the-art methods, which generally use something like transformer decoders with greedy decoding.
> * We have added the comparison with SOTA in the revised version of the paper under Table 1. We also updated the
> results using BERT-large for a fairer comparison with SOTA (we only compare to models using sentence-level context).
> We obtained the best results on the _Conll-2003_ dataset and comparable on _OntoNotes 5.0_.
>
> > The authors might want to speculate on why that is-- it isn't clear to me a priori that the proposed method should be more accurate.
> * We performed an ablation on this model, and found that the introduction of local filtering is the main source of
> improvement since without the global loss, the performances are already very competitive (better than CRF and
> SemiCRF). However, the global loss systematically increases the performance. See the updated Table 1
> and added an ablation section (5.2).
> * We speculate that one reason may be due to the fact that the inductive bias of the Semi-CRF may be too strong and therefore does not allow the model to have much flexibility. Thus, our model adds more flexibility through the local classifier while respecting the structure of the output (non-overlapping segments). We will try to understand this better in future work from a learning dynamics perspective.
>
> > it introduces a hyperparameter beta, it has the possibility of discarding good spans in the filtering phase, and the worst-case complexity is higher.
> * The beta hyperparameter is in fact a regularization term that avoids overconfidence of the local classifier towards
> the null lables since most segments are in fact null (unbalanced classification problem): this hyperparameter enables to
> balance the loss. Thus, the addition of this hyperparameter allows a better recall for the filtering. We have updated
> the article to make this clearer.
>
>
> > For many tasks it is possible to give a reasonable upper bound on the span length that needs to be considered, so complexity of semiCRF training/inference is reduced to linear from quadratic in sentence length.
> * In fact, we already did this in the article to reduce the complexity from $L^2$ to $LK$ where $K$ is the maximum
> segment size. In Section 4 of the previous version of the article, we wrote the following:
> “for the segment-based models, we restrict the segment to a maximum width to reduce complexity without harming the recall score on the training set (however some segments may be missed for the test set).”
> * We updated the section 4 to make this more explicit.
>
> > What about a baseline that uses a maximum segment length and allows null labels only for spans of length 1? I think that should address the weaknesses of semiCRF in a cleaner way and I would not be surprised to see it working quite well in terms of evaluation.
> * Thanks for the suggestion, we added a baseline that allows null labels only for spans of length 1, in Table 1.
> This approach improves performance on Conll-2003 and OntoNotes 5.0 datasets, but the scores are still much
> lower than CRF and FSemiCRF. This result shows that the addition of the local filtering classifier is important for
> improving the performance of the model.

---

> ### Author Response · Authors · 2022-12-08
> **Thank you again for your review**
>
> Dear Reviewer YFd7,
>
> Thank you for your thorough review and valuable feedback on our paper. As mentioned in our previous comments, we have made the suggested changes and believe that they greatly improve the overall quality of the work. We would be grateful if you could take another look and consider increasing your score to reflect the improvements made.
>
> Please note that the response period is ending soon, so if possible, it would be greatly appreciated if you could respond as soon as possible.
>
> Thank you for your time and consideration.
>
> Paper #6275 Authors

---

### Official Review · Reviewer_JeoQ · 2022-10-26

**Confidence:** 3
**Correctness:** 3
**Technical Novelty And Significance:** 2
**Empirical Novelty And Significance:** 3
**Recommendation:** 5

**Clarity, Quality, Novelty And Reproducibility:**

The idea of accelerating the inference of SemiCRF is novel and the proposed filtering method is well-motivated. However, current analysis and experiment results do not support the claim well and more detailed analysis is needed.

**Strength And Weaknesses:**

Strength:
1. Experiment results on performance and throughput verify the claim that Filtered Semi-Markov CRF are better than CRF and Semi-Markov CRF on NER tasks.

Weakness:
1. Some parts of the paper is a little hard to follow, especially Section 3. For example, what is the model architecture of local classifier compared to the global ones?  Do they share parameters? Besides, without reading the rest part in Section 3, it is hard to recognize that Eq. (6) chooses segments whose labels are not predicted to null but also achieves the highest scores. There also exist some typos. For example, in the 4th row from the bottom of Page 3, $j_k<i_{k^*}$, which should be $i_{k^*} < j_k$.

2. Lack of detailed analysis about why filtering leads to better performance. FSemiCRF does Viterbi decoding on a subset of segments compared to the original SemiCRF, whose optimum should be worse than that of SemiCRF in theory. I'm curious whether the introducing of local filtering models and corresponding segment filtering in training or segment filtering during inference is the primary cause of performance improvement. an ablation study and analysis will be helpful. For example, how FSemiCRF performs if it is trained as described in Section 3 but still follows original SemiCRF (i.e., run Viterbi decoding on all the segments) during inference?

3. Model throughput needs further explanation. During inference, both SemiCRF and FSemiCRF have two steps: (1) to calculate segment scores for all segments; (2) Viterbi decoding, in which SemiCRF does Viterbi decoding directly while FSemiCRF runs segment filtering and does Viterbi decoding on the rest segments. It will be better to show which step is the bottleneck (e.g., to show empirical wall-clock time of these two steps) and how FSemiCRF improves SemiCRF in the second step (e.g., to show wall-clock time of segment filtering and Viterbi decoding on the rest segments).

4. What about training complexity (e.g., wall-clock time) of FSemiCRF compared to SemiCRF.

**Summary Of The Paper:**

This paper focuses on reducing the inference complexity of semi-Markov CRF and thus achieving better performance on NER by introducing a filtering step before forward algorithm and Viterbi decoding. More specifically, segments that are predicted to be null and whose label does not achieve the highest predicted score according to an additional local classifier will be filtered. Experiments are conducted on the task of NER and evaluated on three datasets: CONLL-2003, OntoNotes 5.0 and Arabic ACE.

**Summary Of The Review:**

In general, this paper focuses on an important problem w.r.t. SemiCRF which prevents its widespread application as CRF and proposes an applicable method for resolving it. However, this method lacks thorough analysis (especially experiments), which weakens the claims made in the paper.

---

> ### Author Response · Authors · 2022-11-18
> **Response to Reviewer JeoQ**
>
> Thank you for your review. Here are our response:
>
> > Some parts of the paper are a little hard to follow, especially Section 3.
>
> * We revised Section 3 by dividing it into three parts: 1) Filtering, 2) Scoring, training and inference,
> and 3) Complexity analysis. We have also corrected typos and added details to clarify confusion.
> * We have also added Figure 1 which illustrates the different steps of the filtered Semi-CRF.
>
> > what is the model architecture of local classifier compared to the global ones? Do they share parameters?
> * We provide this in the section 4.1 of the paper (**Segment scores** paragraph)
> * The local and global classifiers are two linear layers that takes as input the segment representations, where the representations of the segment $(i, j)$ is computed as $$s_{i:j} = \texttt{SUM}([h_i, h_{i+1}, \dots, h_j])$$
>
> > FSemiCRF does Viterbi decoding on a subset of segments compared to the original SemiCRF, whose optimum should be worse
>  than that of SemiCRF in theory. I'm curious whether the introduction of local filtering models and corresponding
>  segment filtering in training or segment filtering during inference is the primary cause of performance improvement.
>  an ablation study and analysis will be helpful.
>  * Thank you for this suggestion. We performed an ablation of the Semi-Filtered CRF by removing the global loss, i.e.
> keeping only the filtering loss: we have updated Table 1 and added an ablation section (5.2). We found that,
> indeed, the introduction of local filtering models is the principal cause of the performance improvement,
> since without the global loss, the performance is already very competitive. However, having the global
> loss consistently increases the performance.
> * We speculate that one reason may be due to the fact that the inductive bias of the Semi-CRF may be too strong and therefore does not allow the model to have much flexibility. Thus, our model adds more flexibility through the local classifier while respecting the structure of the output (non-overlapping segments). We will try to understand this better in future work from a learning dynamics perspective.
>
> >how FSemiCRF performs if it is trained as described in Section 3 but still follows original SemiCRF
> > (i.e., run Viterbi decoding on all the segments) during inference
> * It is actually not possible to run Viterbi decoding on all segments because our global classifier does not have
> label scores for null labels since they are filtered out. Thus, running Viterbi on all segments will automatically
> lead to bad results.
>
> > It will be better to show which step is the bottleneck (e.g., to show empirical wall-clock time of these two steps)
> and how FSemiCRF improves SemiCRF in the second step (e.g., to show wall-clock time of segment filtering and
> Viterbi decoding on the rest segments).
> > What about training complexity (e.g., wall-clock time) of FSemiCRF compared to SemiCRF.
> * We have added a wall clock time in Table 3, which reports the time for segment scoring (bert + space representation
> score calculation) and decoding. We report the results for both Semi-CRF and FSemiCRF. This shows that filtering
> significantly accelerate the decoding process.

---

> ### Author Response · Authors · 2022-12-08
> **Thank you again for your review**
>
> Dear Reviewer,
>
> Thank you for reviewing our paper. We have made significant improvements based on your feedback, including addressing your concern about the complexity of the model and we tried to improve the overall clarity of our paper. Please see the revised version for details.
>
> We would greatly appreciate it if you could take a look at the revised paper and let us know if you have any further comments or suggestions. Please note that the discussion period is ending soon, so we would be grateful for a timely response.
>
> Thank you again for your time and expertise.
>
> Best regards,
> Paper #6275 Authors

---

### Official Review · Reviewer_z7Y1 · 2022-11-07

**Confidence:** 5
**Correctness:** 2
**Technical Novelty And Significance:** 2
**Empirical Novelty And Significance:** 2
**Recommendation:** 5

**Clarity, Quality, Novelty And Reproducibility:**

The clarity (and consequently the reproducibility) could be improved (see remark above). The originality is difficult to judge given that the details of the contribution are missing.

**Strength And Weaknesses:**

Strengths

Attempt to learn segmentation jointly with classification (here named entity recognition or NER) that does not rely on an expansion of the label space (as is commonly done in NER by specializing the labels into begin label, intermediate and outside label).

Weaknesses

-	The model relies on an extra classifier that filters segments. The model is jointly trained by considering the filter classification loss, the segmentation loss and the NER classification loss. It is not clear from the paper how during training the computational complexity is reduced as initially all potential segments need to be considered in the inference step during training. That means that the complexity is equal to the complexity of the semi-Markov model.  The authors acknowledge that “during training, especially in the first stage, the graph size can be large since the filtering is poor.” It is not clear how exactly the filtering is performed during training.
-	Because the segment filtering is the contribution of the paper, its approach (which should be explained in detail) should be separately evaluated, that is, its behavior and reduction in complexity during training and ablating the assumptions made here.
-	Results could have been evaluated on many more NER datasets.
-	The claim that the proposed filtering model drastically reduces the search space compared to a CRF and SemiCRF model is not seen in the model’s throughput numbers, especially not during training. Moreover, the effect of dynamic programming on the model’s throughput during inference on the test data could be investigated. Is dynamic programming also used in the decoding of the CRF and SemiCRF models?


**Summary Of The Paper:**

The paper proposes a semi-Markov conditional random field model that integrates a filtering step to eliminate irrelevant segments when performing named entity recognition in text. According to the authors this helps reducing the complexity compared to a semi-Markov CRF and dramatically reduces the search space.


**Summary Of The Review:**

The paper is below the threshold for acceptance at ICLR because of:
- Important details and evaluations are missing.

Several of my questions were answered during the rebuttal, for which I thank the authors.

---

> ### Author Response · Authors · 2022-11-18
> **Response to Reviewer z7Y1**
>
> Thank you for your review. Here are our response to your concerns:
>
> > It is not clear from the paper how during training the computational complexity is reduced as initially all
> > potential segments need to be considered in the inference step during training. That means that the complexity is
> > equal to the complexity of the semi-Markov model.
>
> Admittedly, the calculation of segment scores is identical for Semi-CRF and FSemi-CRF, but this step can be performed
> efficiently since the segment scores are independent of each other, i.e., they can be calculated in parallel. However,
> the complexity is not the same for decoding, which is significantly slower for Semi-CRF.
>
> To support our claim, we have added "Table 3" which reports the comparison of the average wall-clock time of the Semi-CRF
> and the filtered Semi-CRF, by separating the scoring (bert+span representation) and the decoding.
> This table shows that decoding is significantly longer for Semi-CRF.
>
> > The authors acknowledge that “during training, especially in the first stage, the graph size can be large
> > since the filtering is poor.”
>
> That's right, we added Figure 3 "Evolution of Graph Size During Training" to illustrate this point.
> The plots show that there are many more segments at the beginning of the training, which is normal since the filtering
> model is not yet trained.
>
> > It is not clear how exactly the filtering is performed during training.
>
> We perform filtering using equation 6 and constraining the gold path/segmentation to be a path of the graph.
> We have added Figure 1 to illustrate the filtering process. We also provide more details in the "Training" paragraph
> of section 3.2.
>
> > Because the segment filtering is the contribution of the paper, its approach (which should be explained in detail)
> should be separately evaluated, that is, its behavior and reduction in complexity during training and ablating
> the assumptions made here.
>
> * The filtering actually discards the segments that were predicted to be "null" (i.e., non-entity segments in the case
> of NER) by the local classifier $\phi_{local}$ (see equation 6).
>
> * We provide an ablation on Table 1, which shows the effect of filtering on model performance by removing the global
> loss of FSemiCRF.
>
> * Table 3 also reports contribution of each component to the wall clock (score calculation and decoding algorithm
> execution time). This shows that filtering significantly accelerate the decoding process.
>
> > Results could have been evaluated on many more NER datasets.
>
> Unfortunately, our algorithm is only applicable to flat NER data (i.e., without nested or overlapping segments),
> making it inapplicable to other datasets.
>
> > Is dynamic programming also used in the decoding of the CRF and SemiCRF models?
>
> Indeed, the CRF and Semi-CRF use forward/viterbi algorithms that fall under dynamic programming. The algorithms are
> provided in the Appendix (A2 and A3).

---

> ### Author Response · Authors · 2022-11-29
> **Thank you Reviewer z7Y1**
>
> We are profoundly grateful to you for increasing your recommendation score and helping us to improve the quality of our paper. If you have any further questions/concerns, we will be happy to answer them, especially about the "correctness" of our article for which you keep a low score.

---

### Author Response · Authors · 2022-11-18
**General response to the reviewers**

We are very grateful to the reviewers for their insightful comments that helped us improve the paper.

Here is the list of changes to address the concerns of the reviewers:
* We revised Section 3 by dividing it into three parts: 1) Filtering, 2) Scoring, training and inference,
and 3) Complexity analysis. We have corrected the typos and added details to clarify confusion. We have also added Figure 1 which illustrates the different steps of the filtered Semi-CRF.
* We added some clarification in the introduction
* We move the proofs of propositions 2 and 3 to the appendix.
* We added Figure 3, which plots the evolution of the graph size during training, to address the reviewer's comment
about training throughput. It confirms our assertion that graph size is large at the beginning of training.
* We reported some of the results of SOTA in Table 1. We also reported our experimental results using "bert-large-cased"
instead of "bert-base-cased" for a fair comparison with existing models.
* We added two ablation studies at section 5.2:
    - FSemiCRF without global loss to show that local filtering is the main source of improvement.
    - Semi-CRF variant that allows zero labels only for spans of length 1
* We added a "Efficiency analysis" at section 5.3, which analysise the results of the throughput and wall clock time table
* We added a wall clock time in table 3 that reports the comparison of the average wall clock time of the Semi-CRF and
the filtered Semi-CRF in milliseconds (per sample) by separating the time needed to compute the segment representations
(with Bert) and the time needed for decoding.

---

### Author Response · Authors · 2022-11-21
**To reviewers**

Dear Reviewers

All of your early concerns about our article have been addressed. If you believe there are still concerns or queries, we look forward to discussing it with you. We also appreciate any further suggestions.

Best,

Paper #6275 Authors

---

### Decision · Program_Chairs · 2023-01-20

**Decision:**

Reject

**Justification For Why Not Higher Score:**

Although the authors have addressed some of the reviewers' comments and improved the paper, the paper still suffers from a lack of clarity regarding the details of the method. The method can only operate on flat NER data, and many transformer-based NER methods can perform well on non-flat data. Furthermore, there are no comparisons with the SOTA.

**Justification For Why Not Lower Score:**

N/A

**Metareview: Summary, Strengths And Weaknesses:**

The paper presents a new method, Filtered Semi-Markov CRF for NER tasks, which addresses the complexity and performance issues of Semi-CRF. Although the authors have addressed some of the reviewers' comments and improved the paper, the paper still suffers from a lack of clarity regarding the details of the method. The method can only operate on flat NER data and there are many transformer-based NER methods that can perform well on non-flat data. Furthermore, there are no comparisons with the SOTA.